# Supporting Heterogenous Traffic on Top of Point-to-Multipoint Light-Trees [note 1]

**DOI:** 10.3390/s23052500

**Published:** 2023-02-23

**Authors:** Masab Iqbal, Luis Velasco, Marc Ruiz, Nelson Costa, Antonio Napoli, Joao Pedro, Jaume Comellas

**Affiliations:** 1Advanced Broadband Communications Center (CCABA), Universitat Politècnica de Catalunya (UPC), 08034 Barcelona, Spain; 2Infinera Unipessoal Lda., 2790-078 Carnaxide, Portugal; 3Infinera, 81541 Munich, Germany; 4Instituto de Telecomunicações, Instituto Superior Técnico, 1049-001 Lisbon, Portugal

**Keywords:** optical point-to-multipoint, multilayer optical networks, metro and access networks

## Abstract

New 5 G and beyond services demand innovative solutions in optical transport to increase efficiency and flexibility and reduce capital (CAPEX) and operational (OPEX) expenditures to support heterogeneous and dynamic traffic. In this context, optical point-to-multipoint (P2MP) connectivity is seen as an alternative to provide connectivity to multiple sites from a single source, thus potentially both reducing CAPEX and OPEX. Digital subcarrier multiplexing (DSCM) has been shown as a feasible candidate for optical P2MP in view of its ability to generate multiple subcarriers (SC) in the frequency domain that can be used to serve several destinations. This paper proposes a different technology, named optical constellation slicing (OCS), that enables a source to communicate with multiple destinations by focusing on the time domain. OCS is described in detail and compared to DSCM by simulation, where the results show that both OCS and DSCM provide a good performance in terms of the bit error rate (BER) for access/metro applications. An exhaustive quantitative study is afterwards carried out to compare OCS and DSCM considering its support to dynamic packet layer P2P traffic only and mixed P2P and P2MP traffic; throughput, efficiency, and cost are used here as the metrics. As a baseline for comparison, the traditional optical P2P solution is also considered in this study. Numerical results show that OCS and DSCM provide a better efficiency and cost savings than traditional optical P2P connectivity. For P2P only traffic, OCS and DSCM are utmost 14.6% more efficient than the traditional lightpath solution, whereas for heterogeneous P2P + P2MP traffic, a 25% efficiency improvement is achieved, making OCS 12% more efficient than DSCM. Interestingly, the results show that for P2P only traffic, DSCM provides more savings of up to 12% than OCS, whereas for heterogeneous traffic, OCS can save up to 24.6% more than DSCM.

## 1. Introduction

The introduction of new services for different use cases, such as augmented reality, virtual reality, internet of skills, industry 4.0 and robotics, etc., ref. [1] will require extending the optical network towards the edges [2] and integrating optical and radio networks [3] which will add more diversity to the requirements for the traffic that the optical transport needs to support. Such requirements include not only data rates, but also varying directionality and traffic patterns, as well as connectivity types, e.g., between one source and one or multiple destinations. Therefore, innovative solutions in optical transport need to be devised that provide the required agility in terms of dynamic traffic management and flexibility in terms of topology upgrade [4].

In this regard, point-to-multipoint (P2MP) networking has been recently proposed at the optical layer to connect several mobile sites to a single metro datacenter [5]. In P2MP connections, one single source node *(hub)* is connected to a set of destinations (*spokes*) that may be scattered over a geographical area. Note, in contrast, that point-to-point (P2P) connectivity has been generally implemented in transport networks, which does not fit well to support *hub-and-spoke* arrangements such as the above one. However, P2MP also has disadvantages, such as the loss of data privacy, since all the leaves can receive all the data being sent.

Implementing P2MP at the optical layer requires not only establishing *light-trees* [6,7], but also innovative optical transmission technologies based on advanced digital signal processing (DSP) [8]. DSP is commonly used to implement high-order modulation formats that increases the speed and reach of coherent optical systems [9]. DSP can also be used to implement new optical communications technologies, such as digital subcarrier multiplexing (DSCM) [10,11]. The primary advantage of DSCM is that it enables fixed bandwidth granularity, i.e., instead of generating a single carrier at the transmitter, several subcarriers (SC) are generated and multiplexed digitally before transmission. Additionally, SCs can be dynamically activated and deactivated to meet capacity requirements [12]. DSCM was recently proposed for P2MP coherent communications in [5,13,14,15], where a subset of independent SCs can be dedicated to support P2P traffic between the source and each destination with the benefit of using one single optical transceiver in the source supporting all the P2P traffic flows.

A different optical communications technology for optical P2MP supporting P2P traffic is optical constellation slicing (OCS) [16]. OCS digitally slices the optical constellation of a single carrier to transport a dynamically defined combination of P2P and P2MP traffic flows. The slicing is performed by dedicating a different subset of constellation points (CP) to different receivers. This paper studies the feasibility of OCS for a dual-polarization (DP), 64 GBd system. We compare the throughput and relative capital (CAPEX) and operational (OPEX) expenditures of OCS and DSCM technologies for different optical P2MP scenarios. Many previous works have compared different solutions in terms of CAPEX and OPEX. For instance, the authors in [17] compared next-generation central offices for metro networks, the authors in [2] envision an optical transport extending from the core to the network edge, and the authors in [18] studied the cost of solutions for a multilayer packet over optical networks.

Additionally, this paper addresses data privacy P2MP connections. To prevent data breach, an independent lookup table (LUT) for each traffic flow is used to implement a substitution cipher [19]. As in our previous paper [20] for securing lightpaths, the LUT is used at the source to encode the transmitted data which also makes sure that only the destination(s) with the corresponding LUT can decode the data that have been assigned to it. Note that although substitution ciphers do not provide perfect secrecy, they provide enough degree of privacy for the type of applications that we target.

The rest of the paper is as follows. Section 2 investigates P2P and P2MP connectivity considering the limits of the existing deployed P2P network and shows how P2MP at the optical layer might support both P2P and P2MP packet traffic and make the transport architecture more efficient. Two optical technologies, DSCM and OCS, are selected as candidates to implement P2MP connectivity using coherent optical communication systems. Section 3 discusses OCS in detail and highlights the construction, implementation, and throughputs of OCS compared to that of DSCM. Section 4 compares the optical performance of OCS and DSCM by simulation and presents numerical results to provide a quantitative comparison between them under dynamic traffic scenarios. Finally, Section 5 draws the main conclusions of the paper.

## 2. Optical Layer Supporting P2P and P2MP Traffic

P2P and P2MP communications are illustrated in Figure 1a,b. Figure 1a presents a particular scenario of P2P connectivity, where all the sources are in the same location and the destinations are in multiple remote locations, i.e., a hub-and-spoke arrangement, which is typical in, e.g., access networks. Figure 1b shows the same scenario for P2MP connectivity, where one single source communicates with destinations in multiple remote locations. Some differences are worth highlighting: (i) from the pure connectivity viewpoint, in a P2P connection, the signal sent by a source is received only by its destination counterpart, while in a P2MP connection, the sourced signal is received by all the destinations; (ii) one single source is used and part of the communication resources can be shared in P2MP connectivity, whereas dedicated sources and connectivity resources are needed in P2P connectivity. Therefore, in the case of *N* destinations, 2× *N terminals* with dedicated connectivity resources are involved in the case of P2P connectivity, whereas only *N* + 1 *terminals* are involved in the case of P2MP connectivity.

P2MP can be easily implemented on the optical layer by creating light-trees connecting one hub node with multiple leaves [7]. However, most services involve P2P traffic and thus the theoretical benefits of P2MP in terms of resource savings can be difficult to be collected in practice. For this very reason, we explore the application of P2MP communications at the optical layer while supporting P2P traffic, and possibly P2MP traffic, at the packet layer (Figure 1c). Here, the number and dimensioning of the optical transceivers are key performance indicators of the cost in terms of CAPEX and OPEX for the network operator compared to the traditional P2P connectivity with dedicated lightpaths and transceivers. In addition, the efficiency of the solution and the simplicity of the architecture should also be considered in the analysis.

Two optical technologies have been selected as they can be used for optical P2MP and support P2P traffic at the packet layer: DSCM and OCS. In the case of DSCM (Figure 2a), the spectrum is separated into a number of Nyquist digital subcarriers (SC), e.g., 16, 4 GBd SCs, and multiplexed together at the hub node to create a 64 GBd signal. The hub node sends the SCs to all the leaves, where lower-capacity transceivers listen in to their assigned frequencies to receive their dedicated SCs. In the example, we represent four destinations all operating a sub-set of four SCs. To simplify the hardware implementation of DSCM, the SCs assigned to each destination must be contiguous in the spectrum, which means that P2MP traffic can be supported only for destinations with neighboring SCs.

In the case of OCS (Figure 2b), a single optical signal is generated at the hub node and sent to the leaves. The optical constellation is sliced, and every slice is used to support P2P or P2MP traffic. In the example in Figure 2b, five optical constellation slices are defined (OCS 1 to 5), where four of them support P2P traffic between the source node and one of the destinations, whereas OCS5 supports P2MP traffic. It is worth noting that the effective bitrate of an OCS can be controlled by selecting the number of CPs that are assigned to it. In the example in Figure 2b, the optical signal is modulated using 64-QAM, so 64 CPs are available. From them, OCS1 is assigned 32 points, whereas OCS 3 and 4 are assigned just 4. Only the CPs assigned to one OCS can be used for communication to the corresponding destination.

It seems clear that implementing optical P2MP using DSCM or OCS can bring benefits in terms of cost, efficiency, and simplicity of architecture. CAPEX reduction comes as a result of decreasing the number of required transceivers. Moreover, OCSs and SCs can be created, modified, and eliminated dynamically, which provides the needed flexibility of the connections as per traffic demands.

## 3. Optical Constellation Slicing

This section details the OCS solution that includes the design of the transmitter and receiver and the theoretical throughput.

Implementing OCS requires adding encoder and decoder blocks to the standard coherent transceivers, as shown in Figure 3. The encoder and decoder blocks are in charge of slicing the optical constellation to support P2P and P2MP traffic on top of the optical P2MP.

For illustrative purposes, Figure 4 shows two examples of alternative configurations that can be supported using OCS on the same light-tree. The example in Figure 4a shows the traffic that can be supported with the configuration presented in Figure 2b, i.e., four P2P traffic flows between the source and every destination, as well as one P2MP traffic flow between the source and all the destinations. Only by configuring the encoder and decoder can the traffic shown in Figure 4b be supported, which entails two P2MP traffic flows, each between the source and two different destinations. These examples are intended to demonstrate the flexibility of OCS to allow dynamic changes in the configuration of the optical system as a function of the traffic requirements, without changing the light-tree itself. To protect data that is not intended to a specific destination, the encoder block implements a substitution cipher for every OCS based on a specific LUT, which encrypts data before transmission, while the decoder implements the inverse operation for every OCS*i* individually.

Figure 5 details the OCS encoder for the configuration in Figure 2b. The first step is to choose the modulation format (*m*) of quadrature amplitude modulation (QAM) that provides the highest throughput and the maximum number of OCSs. Because the lowest number of CPs that may be allocated to a single OCS*i* is two (each point representing a single data bit), the maximum number of simultaneous destinations is *m*/2. For the sake of clarity, we assume *m* = 64 hereafter.

The next step is to create the OCSs. Figure 5 illustrates the traffic shaping to be implemented at the TX side. Every OCS*i* is then associated with a buffer within the TX, where data streams are temporarily stored. From those buffers, sets of bits of size, equal to the number of *bits with information* (*infobits*) in the OCS*i*, are selected and encrypted using the LUT as a substitution cipher. For example, in Figure 5, sets of five bits are selected for OCS1, whereas sets of three bits are selected for OCS2. Next, the shaping block receives one of the encrypted sets from each OCS at a time and adds the prefix that identifies the OCS for which that data is intended; this results in a specific symbol in the optical constellation. Hence, each RX receives symbols formed from a unique <*prefix*, *infobits*> pair, where the prefixes are decided according to the constellation map. Note that prefixes might be of different lengths, so sets of six bits are obtained to feed the 64-QAM optical modulator. For example, it adds prefix 0 to sets of bits from OCS1 and prefix 111 to sets of bits from OCS2 (see Figure 5). The shaping block follows a 64-step cycle, where at every step, it receives a set of bits from an OCS*i*; the number of sets of bits selected from each OCS*i* is exactly the number of CPs assigned to that OCS*i*. For example, the shaping block receives 32 out of 64 5-bit sets from OCS1, and 8 out of 64 3-bit sets from OCS2. Note that it is at the source where most of the functionalities need to be implemented (including slicing and LUT coding), while destinations perform LUT decoding only. With this arrangement, the *throughput* of each OCS*i* can be computed as a function of the number of bits with information, the symbol rate (*SR*), the number of CPs assigned to the OCS*i* (*#CP*), and the modulation format used (*m*-QAM). Note that both gross or net throughput can be computed by considering gross or net *SR* in Equation (1).
(1)$$Throughput_{OCS{}_{i}}=\frac{infobits_{i}}{m}\times \#CP\times SR\;$$

In contrast, Equation (2) can be used to compute the throughput of each SC in a DSCM system working on the same modulation format (*m*), where *N* is the number of SCs.
(2)$$Throughput_{SC}=\frac{log_{2}m}{N}\times SR$$

In both cases, the system’s throughput is calculated as the summation of the individual throughputs. Let us assume *SR* = 64 GBd and 20% FEC overhead for a DP system. In the example in Figure 2b, the throughput of OCS1 and OCS2 is 250 Gb/s and 37.5 Gb/s, respectively. In contrast, for the DSCM system in Figure 2a, the throughput of each SC is 37.5 Gb/s and every RX receives a throughput of 150 Gb/s.

Additionally, for OCS, we define the *slice efficiency* (*SE*) of each slice OCS*i* as the ratio between the number of *infobits* with respect to the total number of bits per symbol (Equation (3)), as well as the *contributed efficiency* (*CE*) as a function *SE* and the number of CPs assigned to the OCS*i* (Equation (4)).
(3)$$SE_{OCS{}_{i}}=\frac{infobits_{i}}{log_{2}m}$$
(4)$$CE_{OCS{}_{i}}=SE_{OCS{}_{i}}\times \frac{\#CP}{m}$$

For illustrative purposes, Table 1 summarizes the maximum throughput, *SE*, and *CE* of each OCS*i* in the system, presented in Figure 2b. For this case, the transceiver provides a 412.5 Gb/s total throughput and reaches a 68.8% efficiency, computed as the summation of the individual throughput and the *CE* of the OCS slices.

## 4. Illustrative Results

In this section, we first compare via simulation the optical performance of OCS and DSCM. Then, to assess the viability of OCS, quantitative analyses are performed for both OCS and DSCM considering the dynamic traffic conditions for two distinct scenarios in which the traffic handled by the light tree is (i) P2P and (ii) P2P + P2MP.

### 4.1. Performance Evaluation of DSCM and OCS for Optical P2MP

For evaluation purposes, we implemented a MATLAB-based simulator. For OCS, a single carrier 64-QAM@64GBd DP OCS signal was generated, whereas for DSCM, we implemented 64-QAM@64GBd DP with 16 SCs each operating at 4 GBd. The signal is then sampled and sent through a root-raised cosine pulse shaper with a roll-off factor of 0.06. For DSCM, after applying frequency shift, the SCs were multiplexed. The signal was launched onto an 80 km long fiber link with *N* spans; an optical amplifier with a noise figure of 4.5 dB compensates for the fiber losses after each span. To simulate ASE noise, additive white Gaussian noise is injected after each span. The fiber channel was simulated using standard single-mode fiber with the following parameters: fiber loss = 0.21 dB/km, dispersion *D* = 16.8 ps/(km-nm), and nonlinear coefficient = 1.14 W^−1^ km^−1^. The payload was generated using 2^13^ pseudorandom symbols. The signal was then transmitted using the symmetric split-step Fourier method [21], which solved the nonlinear Schrödinger equation for signal propagation in the fiber channel. The signal was received coherently, and an ideal chromatic dispersion filter was applied. For DSCM, the signal was first demultiplexed and passed through a matched filter before performing the down sampling.

On this simulation scenario, we first analyze the performance of both OCS and DSCM with 16 SCs. First, Figure 6 depicts the spectra of the single carrier and the DSCM signals, where we observe that both signals use the same bandwidth. To study the performance, we simulated a light-tree with two leaves with 8 and 10 spans, respectively. We vary the launch power in the range [−4, +2] dBm. Figure 7 presents the average bit error rate (BER) of both polarizations as a function of the power for a light-tree with two leaves with 8 and 10 spans, respectively. We observe that DSCM provides a better performance compared to OCS at higher launch powers where non-linearities are significantly present. This is because each SC operates on a reduced baud rate of 4 GBd. An important point to note here is that although for eight spans the performance of DSCM is better than OCS, for optimal power, the BER remains lower than the FEC threshold of 1 × 10^−2^ in both the cases and the post-FEC performance of both systems remains the same. Therefore, if the main goal is to reduce non-linearities, i.e., for long haul systems, DSCM is a superior solution.

Complementing Figure 7, Figure 8 details the optical performance of OCS for each individual polarization (Figure 8a) and the performance of each SC of the DSCM system (Figure 8b), considering for eight spans. The BER of both polarizations are almost the same in OCS. However, the performance of each SC at −1 dbm reveals that the SCs in the center are affected more than the SCs at the extremes because of the SC interaction mainly due to the non-linearities. However, the BER remains under the FEC threshold, thus guaranteeing the same post-FEC performance of each SC.

### 4.2. Quantitative Analysis

Let us first describe the scenario that we consider to analyze the performance of OCS and DSCM and compare it with a lightpath solution that uses traditional optical P2P coherent transceivers. We consider two types of traffic for 2 and 4 destinations: (i) P2P only and (ii) both P2P and P2MP. For the study, the performance metrics analyzed are the number of required transceivers, the cost of deployment, and the efficiency.

For the traditional P2P lightpath solution, the optical transceivers considered are 25 G, 50 G, 100 G, 200 G, and 400 G [22], whereas for OCS and DSCM technologies, we assume 600 G 64-QAM@64GBd DP transceivers, as introduced in Section 4.1. Because in DSCM, receivers might have limitations in terms of the number of SCs they can process, two different availabilities are considered: (i) *non-aggressive*, which corresponds to the current status of P2MP transceivers (e.g., [5]), where transmitters support 16 SCs for a total 600 G capacity, whereas receivers support 8 or 4 SCs for a total capacity of 300 G and 150 G, respectively, and (ii) *aggressive*, where both the sender and receivers have 600 G installed, which enables all the possible configurations. The summary of the configurations to be used in the rest of the paper are summarized in Table 2.

Regarding traffic, we consider dynamic traffic scenarios where the capacity requested by destinations varies over time. In this case, a destination needs to support the maximum traffic of the traffic profile. Hence, destinations must require installing the optical transceiver that supports such a data rate or it must possess the capacity to dynamically manage the resource allocation according to the traffic profile. Figure 9a,b presents the traffic profiles under consideration for two and four destinations, respectively, for P2P traffic only and mixed P2P + P2MP traffic. We observe that P2P traffic varies over the day [23], with some destinations requiring more capacity during day light (8 h–18 h), whereas other require more traffic during the night (18 h–8 h). As for the P2MP traffic, we assumed a constant 100 Gb/s capacity throughout the day. The total maximum capacity requirement in the pure P2P traffic scenario is 300 Gb/s, whereas in the mixed one it is 400 Gb/s.

Considering the same traffic profile and transceivers’ availability as described previously, Figure 10 represents savings in terms of the number of transceivers required for OCS and DSCM as compared to the P2P lightpath solution. For the P2P traffic scenario, both OCS and DSCM provide equal savings in both aggressive and non-aggressive availabilities. For 2 destinations, the reduction in transceivers is 25%, whereas for 4 destinations, it is 38%. However, for the P2P + P2MP traffic scenario, we observe a different trend. In the aggressive availability, OCS and DSCM provide the same savings, i.e., 57% and 62% for 2 and 4 destinations, respectively. However, in non-aggressive availability, OCS outperforms DSCM, which provides 14% and 23% savings for 2 and 4 destinations, respectively, in comparison to 57% and 62% of OCS, i.e., 43% and 39% more savings, respectively. The reason behind these results is the additional number of transceivers required in DSCM to serve P2MP traffic when the highest data rate transceiver type is not available at the leaves.

Let us now analyze the optimal cost of OCS and DSCM transceivers to find where they start to provide cost savings and compare with the P2P lightpath solution. To this goal, we model the cost of P2P transceivers and apply the needed constraints to find the targeted cost of OCS and DSCM ones.

For the P2P lightpath solution, we consider the optical transceivers availability as in Table 2. Based on the cost model in [24], we model the cost (*C*) of these transceivers in monetary units (m.u.) as a function of the bit rate (*BR*), as:(5)$$C=\alpha \cdot BR^{\beta },$$
where α is a normalization factor to set the cost of the 100 G transceiver to 1 and *β* is a positive constant < 1 that we use to define different cost profiles. Two cost profiles are considered for analysis, *β* = 0.5 and *β* = 0.6 (see Figure 11), which reproduces the technology costs trend to exponentially decrease by generations [24].

However, to enable optical P2MP connectivity, DSCM-600, DSCM-300, DSCM-150, and OCS-600 transceivers require additional DSP features, which translates into an additional cost. Although the cost of such transceivers still needs to be determined, it is worth noting that we imposed these relations among costs following the proportions in Figure 11, for *β* = 0.5 and *β* = 0.6.

In the aggressive availability (Figure 12a), DSCM outperforms OCS for P2P traffic. For 2 destinations, DSCM provides 12% and 9% extra cost savings than OCS for *β* = 0.5 and *β* = 0.6, respectively. The cost savings are more evident for 4 destinations, where DSCM provides 35.4% and 34% more cost savings than OCS. However, both OCS and DSCM provide the same savings in the case of mixed P2P + P2MP traffic. For *β* = 0.5, we observe savings of 40% and 32% for 2 and 4 destinations, respectively, similarly as for *β* = 0.6, where the savings are 42.2% and 31.4%, respectively. In the non-aggressive availability (Figure 12b), the same savings as for aggressive availability for P2P only traffic are observed because the installed transceivers remain the same in both the cases. However, for mixed P2P + P2MP traffic, OCS outperforms DSCM mainly because additional transceivers for enabling P2MP traffic need to be installed. For 2 destinations, OCS provides 24.6% and 25.8% more cost savings than DSCM for *β* = 0.5 and *β* = 0.6, respectively. The cost savings are lower for 4 destinations, where OCS provides 12.9% and 14.5% more cost savings than DSCM.

Finally, we study the efficiency of DSCM and OCS for optical P2MP, as this metric provides us with an insight into how efficiently a source can utilize the total capacity to serve the requested dynamic traffic profile. For example, in Figure 9a at 8 h, destination 1 requires 250 Gb/s and destination 2 requires 100 Gb/s. In this case, the efficiency of OCS and DSCM will be 58.33% (350/600), whereas for the P2P lightpath solution, the efficiency will be 43.75% (350/800). Figure 13 shows the efficiency for the case of two destinations for the aggressive and non-aggressive availabilities for P2P (Figure 13a) and P2P + P2MP (Figure 13b) traffic profiles. In the case of the aggressive availability, the efficiency of OCS and DSCM is the same throughout the traffic profiles, whereas the efficiency of the P2P lightpath solution is always under that of OCS and DSCM. Under P2P only traffic, the OCS and DSCM approaches are utmost 14.6% more efficient than the P2P lightpath solution that is observed at 8 h. At that same hour of the day, the maximum efficiency improvement of 25% for P2P + P2MP traffic is observed. In the case of the non-aggressive availability, the same pattern as the aggressive availability is observed for P2P only traffic. However, for mixed P2P + P2MP traffic, OCS provides a abetter efficiency than DSCM and achieves a maximum efficiency improvement of 12% at 8 h.

In view of these results, let us examine implementing OCS on top of DSCM. For illustrating the OCS + DSCM solution, we consider a destination that is only operating on 1 SC of the DSCM system with a 150 Gb/s capacity. Suppose that the capacity required by this node reduces over time and more nodes are needed to be installed. With the OCS on top of DSCM solution, the constellation can be sliced and more nodes can be added, allowing for topology reconfigurability. For example, instead of a single node, two nodes with a capacity of 62.5 Gb/s can be installed if 32 CPs are assigned to each node. Similarly, 4 nodes can be enabled by 16 CPs with capacity of 25 Gb/s each, as illustrated in Figure 14.

Furthermore, if the node’s desired capacity is less than the overall capacity of the destination, the capacity sits unused. We can make better use of the destination capacity by adding more nodes and implementing OCS on top of DSCM. Consider the case as shown in Figure 15 where the capacity requested by a destination working on a 150 G SC decreases to 62.5 G. The efficiency of the destination in that case will be 42% (62.5/150). On the other hand, OCS can provide the benefit of adding one more node by performing 32-CP slicing to increase the capacity utilization to 83.33%.

## 5. Conclusions

The optical P2MP is compared against traditional P2P when dealing with dynamic P2P and P2P + P2MP traffic. Two optical P2MP technologies, OCS and DSCM, are studied and compared.

The simulation findings suggest that DSCM decreases non-linearities and outperforms OCS for long-haul applications. However, in access and metro applications, where non-linearities are assumed to be low, the optical performance of OCS and DSCM is similar.

For the sake of a quantitative comparison, aggressive and non-aggressive transceiver availabilities are defined. In the case of pure P2P traffic, both OSC and DSCM greatly reduce the number of transceivers in the network by the same amount, between 25% and 38%, while simplifying the network architecture. DSCM is more cost effective than OCS under both transceiver availabilities. However, when considering P2P + P2MP traffic under the non-aggressive approach, OCS is able to serve them simultaneously without the need for additional transceivers, whereas DSCM requires around 40% more transceivers. In this scenario, the costs savings in OCS are noticeably larger than those in DSCM. Finally, it is worth noting that OCS and DSCM can co-exist and that this combination can be exploited to develop further efficiency improvements.

## Figures and Tables

**Figure 1 sensors-23-02500-f001:**
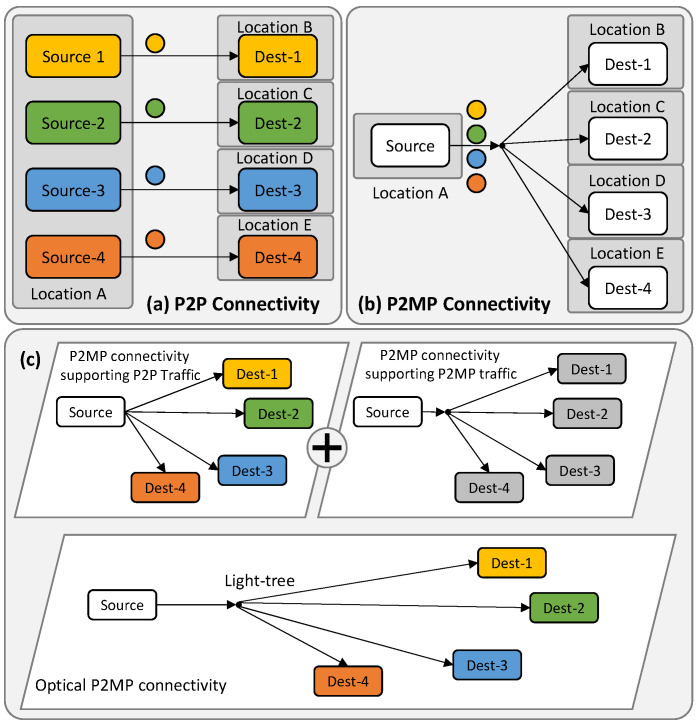
P2P (**a**) and P2MP (**b**) connectivity. Supporting P2P and P2MP traffic on top of optical P2MP (**c**).

**Figure 2 sensors-23-02500-f002:**
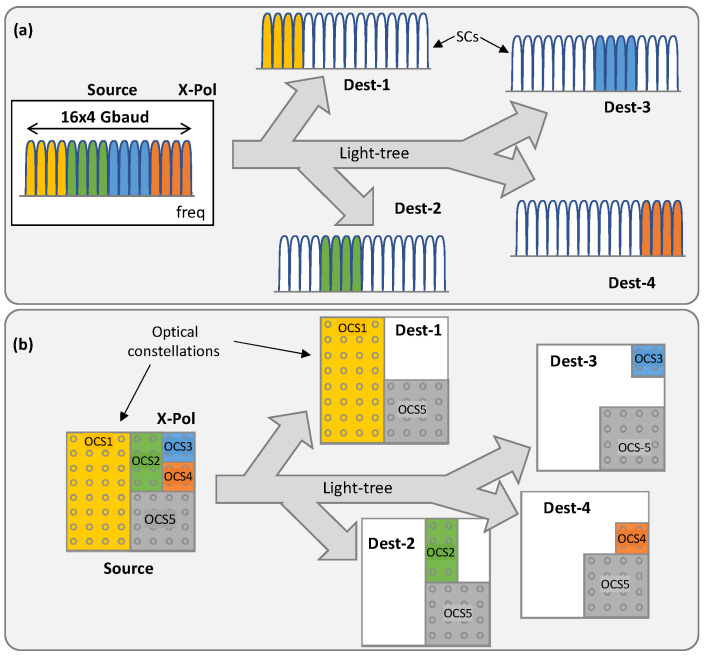
Application of DSCM (**a**) and OCS (**b**) for optical P2MP (updated from [16]).

**Figure 3 sensors-23-02500-f003:**
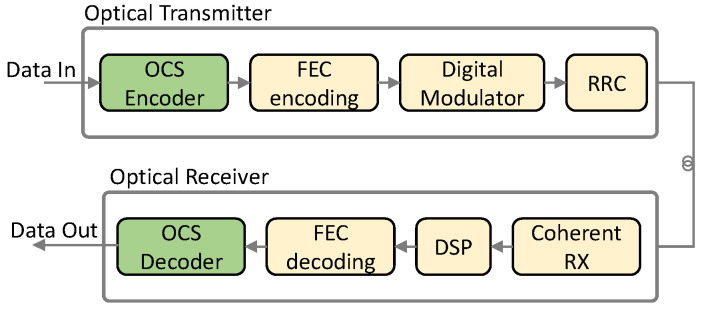
Optical communication system for OCS.

**Figure 4 sensors-23-02500-f004:**
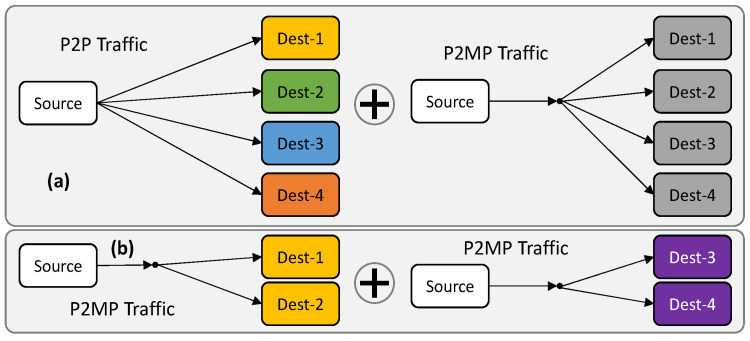
Example of traffic configurations supported by OCS. P2P + P2MP traffic (**a**) and P2MP traffic (**b**) (updated from [16]).

**Figure 5 sensors-23-02500-f005:**
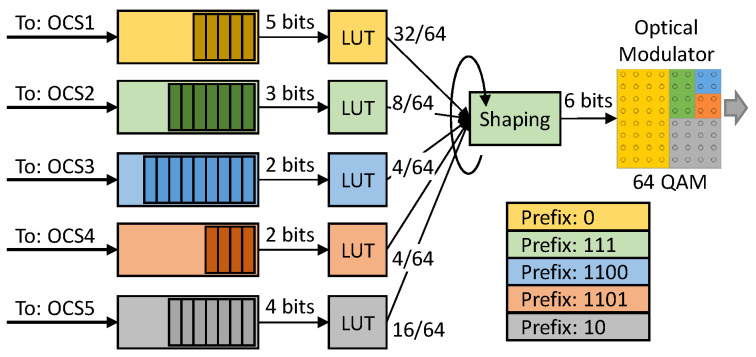
Example of slicing and individual data encryption (updated from [16]).

**Figure 6 sensors-23-02500-f006:**
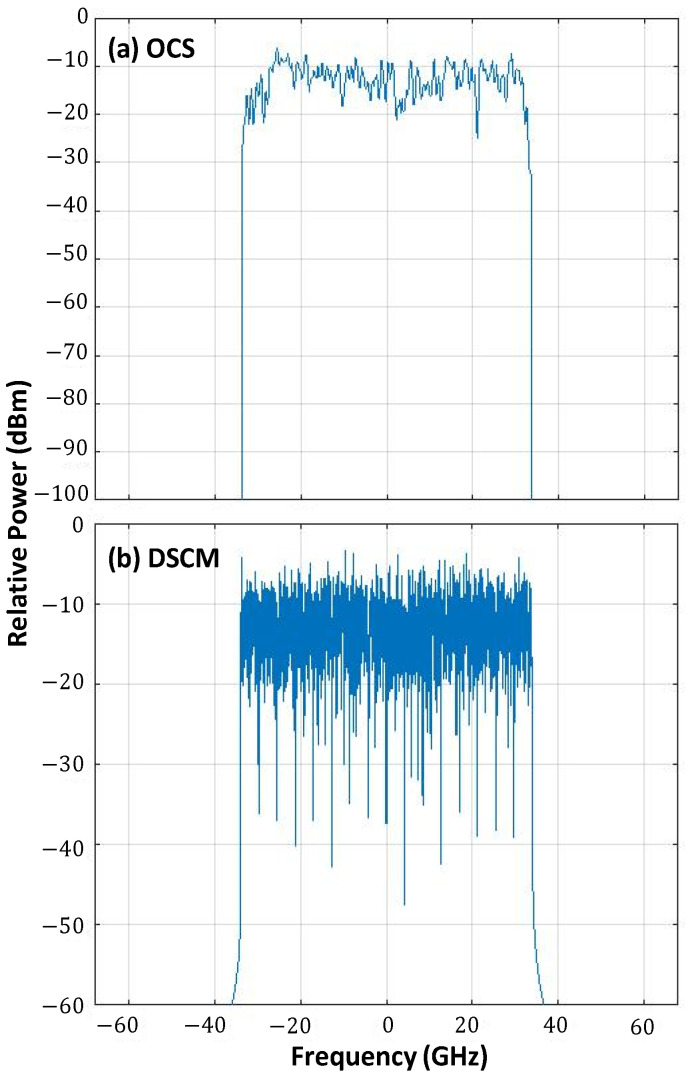
Spectrum of the single carrier signal used for OCS (**a**) and 16 SC DSCM (**b**).

**Figure 7 sensors-23-02500-f007:**
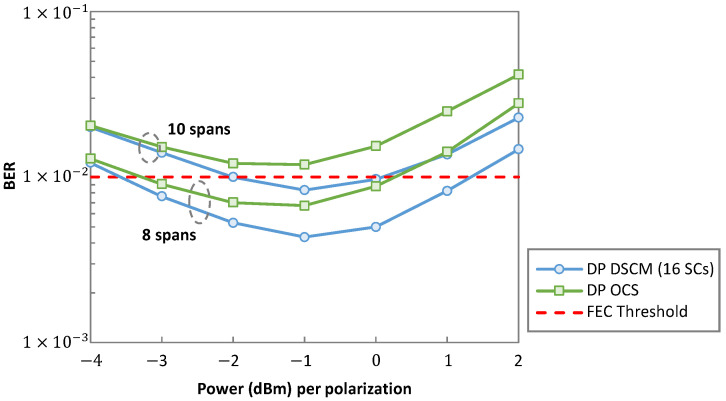
Optical system performance of OCS and DSCM for P2MP.

**Figure 8 sensors-23-02500-f008:**
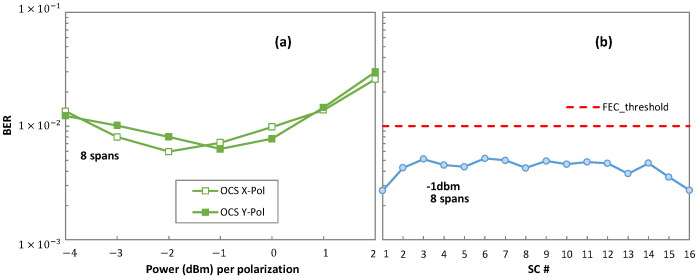
Details of optical performance of OCS (**a**) and DSCM (**b**).

**Figure 9 sensors-23-02500-f009:**
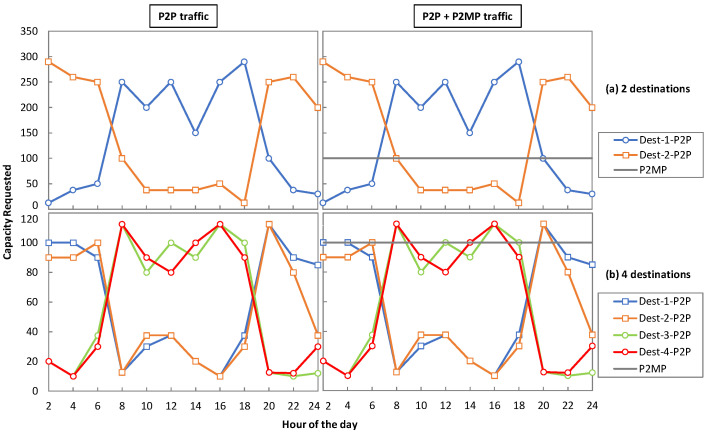
Traffic profiles for P2P and P2P + P2MP traffic with 2 and 4 destinations.

**Figure 10 sensors-23-02500-f010:**
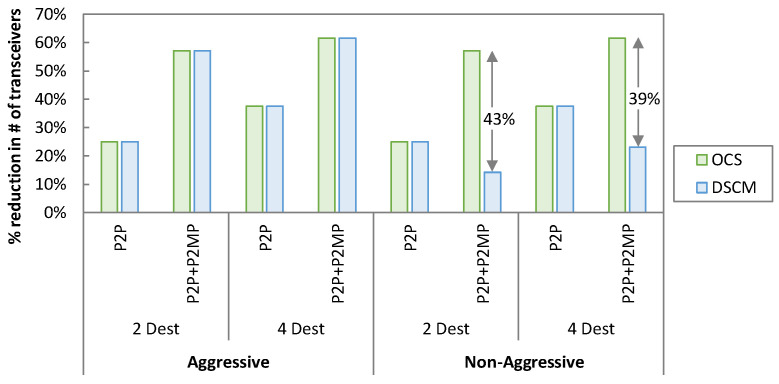
Reduction in number of transceivers.

**Figure 11 sensors-23-02500-f011:**
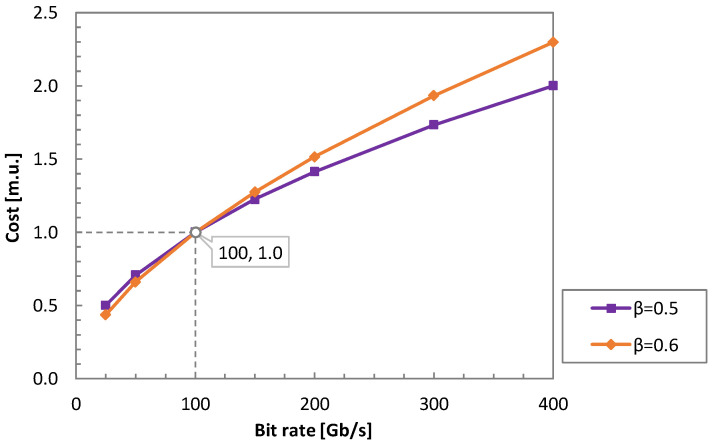
Cost profile for lightpath transceivers.

**Figure 12 sensors-23-02500-f012:**
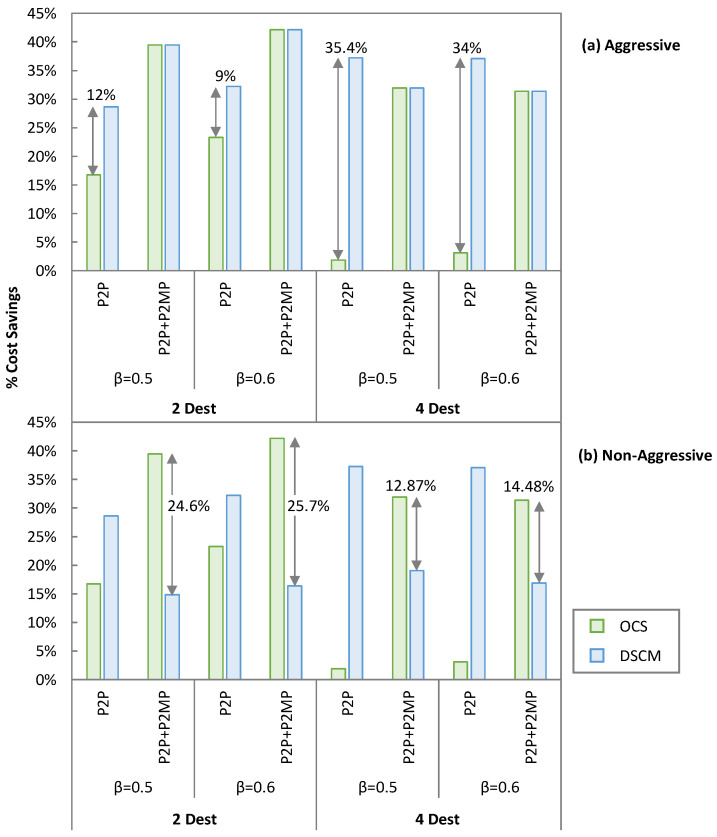
Cost savings: aggressive approach (**a**) and non-aggressive approach (**b**).

**Figure 13 sensors-23-02500-f013:**
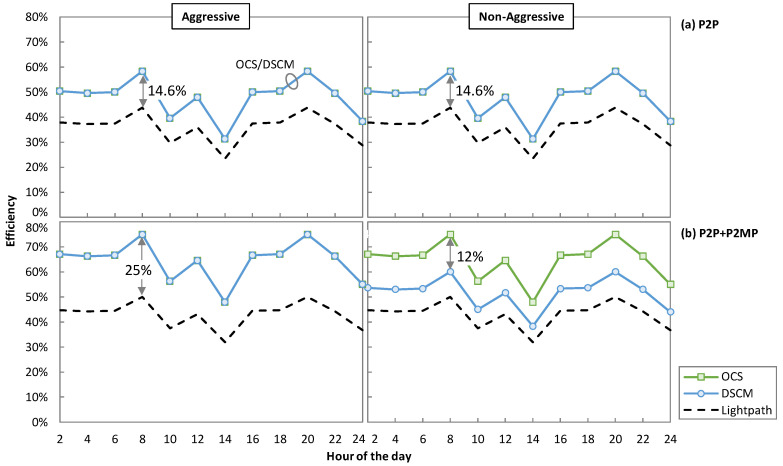
Efficiency of aggressive and non-aggressive availabilities for P2P (**a**) and P2P + P2MP (**b**) traffics.

**Figure 14 sensors-23-02500-f014:**
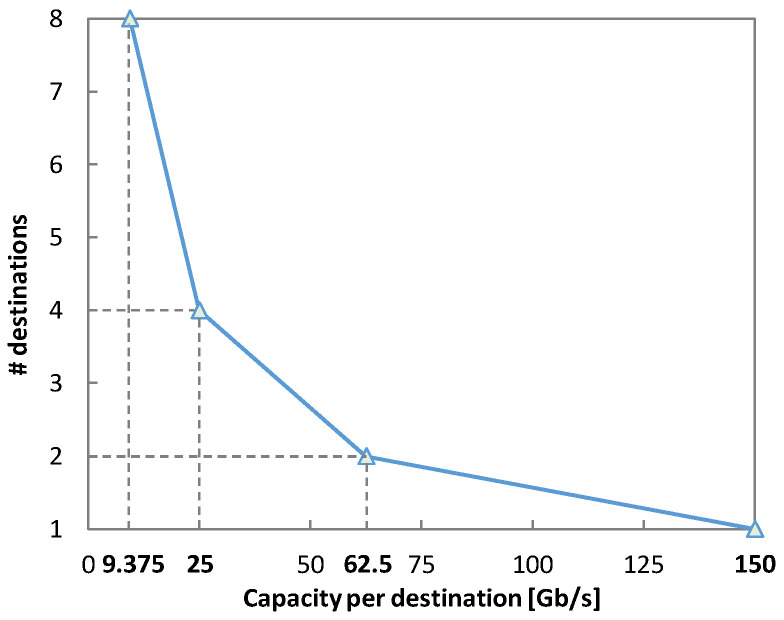
Reconfigurability options in OCS + DSCM.

**Figure 15 sensors-23-02500-f015:**
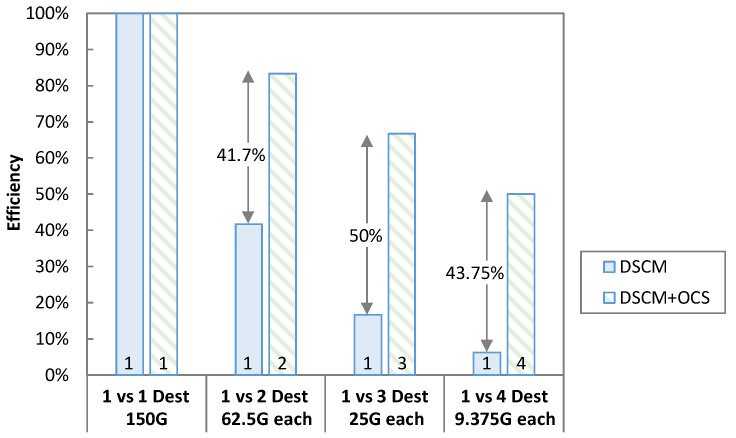
Efficiency improvement for OCS + DSCM.

**Table 1 sensors-23-02500-t001:** Example of throughput, SE, and CE of each OCS_i_.

OCS #	#CP	Throughput [Gb/s]	SE [%]	CE [%]
1	32	250	83.3	41.7
2	8	37.5	50.0	6.3
3	4	12.5	33.3	2.1
4	4	12.5	33.3	2.1
5	16	100	66.7	16.7
Total		412.5	--	68.80

**Table 2 sensors-23-02500-t002:** Available transceivers for P2P, OCS, and DSCM.

P2P	OCS	DSCM	
25 G 50 G 100 G 200 G 400 G	600 G 64-QAM @64GBd DP	600 G 64-QAM @64GBd DP	150 G 4 SCs per polarization 300 G 8 SCs per polarization 600 G 16 SCs per polarization

## Data Availability

Not applicable.

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
