# Peer review of "Supporting Heterogenous Traffic on Top of Point-to-Multipoint Light-Trees†"

_sensors, 2023, doi:10.3390/s23052500_

Round 1

Reviewer 1 Report

P2MP is one of the important issues in the optical network. The paper proposes an Optical Constellation Slicing (OCS) enables a source to communicate with multiple destinations by focusing on the time domain which is described by simulation. The results show that both OCS and DSCM provide good performance for access/metro applications and provide better efficiency and cost savings than traditional optical P2P connectivity. This paper is interesting and good work for optical network especially for P2MP and traffic. However, some comments below which I recommend to give one chance to take a revision. The proposal should give more clear in the compared to previous work such as resource allocation with edge-cloud collaborative traffic prediction in integrated radio and optical networks. The collision should consider in P2MP and optica device. 

Reviewer 2 Report

An interesting paper entitled "Supporting Heterogenous Traffic on top of Point-to-Multipoint Light-Trees."

The authors have presented an appreciable concept that holds an appreciable promise for the existing and the emerging 5G radio access technology use case and/or application by targets.

Correct the typos and/or grammatical errors please.

Abstract: The authors should ensure that the acronyms are consistent e.g., "P2PM" is interchanged with "P2MP." The abstract should contain compelling qualitative and quantitative performance metrics that clearly show the key research findings.

Fig. 1. Supporting P2P and P2MP traffic on top of optical P2MP (c) should be revised to depict the correct P2P connectivity. How do you differentiate homogeneous from heterogeneous destinations?

The authors need to explain the physical layer requirements of the propsoed system for multiple input multiple output connectivity applications. 

The authors should rigorously explain the impact of the proposed model on the modulation error ratio of the optical signal with respect to the bounded media. How would this change between vertical and horizontal polarisations of the signal? signals? For each constellation point, what are the effects of changes in the carrier link margins on the signal quality? How would this change for a given data link margin of a satellite communication link?

If the modulation error ratio of each sub-carrier signal changes, how does this affect the system-level throughput and efficiency of the proposed system?

The conclusion should contain compelling qualitative and quantitative performance metrics that clearly show the key research findings.
